# Dipper throated optimization with deep convolutional neural network-based crop classification for remote sensing image analysis

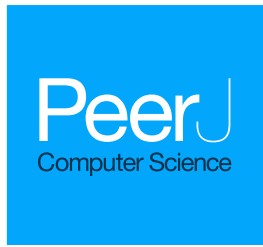

Youseef Alotaibi[1], Brindha Rajendran[2], Geetha Rani K.[3] and Surendran Rajendran[4]

[1] College of Computer and Information Systems, Umm Al Qura University, Makkah, Saudi Arabia
[2] Department of Computing Technologies, SRM Institute of Science and Technology, Kattankulathur, India
[3] Department of Computer Science and Engineering, Jain (Deemed-to-be University), Bangalore, India
[4] Saveetha School of Engineering, Saveetha Institute of Medical and Technical Sciences, Chennai, India

Corresponding author
Surendran Rajendran,
surendranr.sse@saveetha.com

## ABSTRACT

**Problem:** With the rapid advancement of remote sensing technology is that the need for efficient and accurate crop classification methods has become increasingly important. This is due to the ever-growing demand for food security and environmental monitoring. Traditional crop classification methods have limitations in terms of accuracy and scalability, especially when dealing with large datasets of high-resolution remote sensing images. This study aims to develop a novel crop classification technique, named Dipper Throated Optimization with Deep Convolutional Neural Networks based Crop Classification (DTODCNN-CC) for analyzing remote sensing images. The objective is to achieve high classification accuracy for various food crops.

**Methods:** The proposed DTODCNN-CC approach consists of the following key components. Deep convolutional neural network (DCNN) a GoogleNet architecture is employed to extract robust feature vectors from the remote sensing images. The Dipper throated optimization (DTO) optimizer is used for hyper parameter tuning of the GoogleNet model to achieve optimal feature extraction performance. Extreme Learning Machine (ELM): This machine learning algorithm is utilized for the classification of different food crops based on the extracted features. The modified sine cosine algorithm (MSCA) optimization technique is used to fine-tune the parameters of ELM for improved classification accuracy.

**Results:** Extensive experimental analyses are conducted to evaluate the performance of the proposed DTODCNN-CC approach. The results demonstrate that DTODCNN-CC can achieve significantly higher crop classification accuracy compared to other state-of-the-art deep learning methods.

**Conclusion:** The proposed DTODCNN-CC technique provides a promising solution for efficient and accurate crop classification using remote sensing images. This approach has the potential to be a valuable tool for various applications in agriculture, food security, and environmental monitoring.

# INTRODUCTION

Remote sensing image (RSI) classification technology is an essential technology of remote sensing (RS) research, initially; its classification could be explained by artificial visualization (*Joshi et al., 2023*). In the scientific community, computer automatic classification technology is increasingly developed into the mainstream because of its higher-cost labor and great uncertainty (*Eftekhari & Yang, 2023*). It can be categorized into unsupervised and supervised classifications, to enhance the classification accuracy, researchers have recently developed the decision tree, fuzzy mathematics, neural network, and other new techniques (*Mathur, 2021*). With an improvement of higher-resolution RSIs, the necessities of classification are always attained higher, the classification of complex scenes and the needs of rotation translation invariance provide a begin to several researches on deep learning (DL) in the domain of RS Land cover and crop category maps are major necessary inputs while handling with agricultural and environmental monitoring tasks (*Bouguettaya et al., 2023*). Multi-temporal multisource satellite images are commonly needed for capturing particular crop development phases and therefore, capable of discriminating various kinds of crops (*Farmonov et al., 2023*). For instance, multispectral optical images may not be sufficient for separating summertime crops in a heterogeneous and complex environment. Higher-resolution RSIs are extensively employed for classifying crops.

Traditional RSI classification methods, such as statistical approaches and machine learning algorithms, are often limited in their ability to handle the high dimensionality and complex spectral characteristics of RSIs. Additionally, these methods often require extensive manual work for feature extraction and parameter tuning, which can be time-consuming and prone to errors. Effective feature extraction GoogleNet architecture is employed within the DTODCNN-CC framework to extract robust and informative features from RSIs. Standard crop classification techniques mainly based on machine learning (ML), namely, SVM, RF, and KNN, frequently need predetermined features, and the classification outcomes need additional processing (*Bouguettaya et al., 2022*). The classification model is very complex, classification accurateness is lesser, and complex spatial as well as temporal data of higher-resolution RSIs are not proficiently employed (*Tian et al., 2021*). In recent years, the most commonly utilized and effective methodologies for multitemporal and multisensor land cover classification have been ensemble-based and DL methods. These methods are determined to implement the SVM. DL has a robust ML technique to solve an extensive number of tasks occurring in computer vision (CV), natural language processing (NLC), image processing, and signal processing (*Tripathi, Tiwari & Tiwari, 2022*). The major concept is to reproduce the human vision for dealing with big data issues, utilize every data accessible, and offer semantic data as the output. Several techniques, models, and benchmark datasets of reference images are

obtainable in the image classification field (*Suchi et al., 2021*). Recently, many researchers have utilized the DL method for processing RSI. DL approach is shown that effective in processing both optical (multispectral and hyperspectral images) and radar images for extracting various kinds of land cover, namely, building extraction, and road extraction (*Virnodkar et al., 2020*). Current RSI classification methods often suffer from low accuracy when dealing with complex spectral characteristics and high-dimensional data. This significantly limits their effectiveness for real-world applications such as precision agriculture and crop yield estimation. Existing deep learning models offer promising solutions but are prone to overfitting and computationally expensive, hindering their practical applicability. Additionally, the hyper parameter tuning process in these models is often manual and time-consuming, requiring expertise and limiting their accessibility.

This study presents a dipper throated optimization with deep convolutional neural networks based crop classification (DTODCNN-CC) technique for RSI analysis. The presented DTODCNN-CC methodology concentrates on the detection and classification of food crops that exist in RSIs. To accomplish this, DTODCNN-CC technique applies DCNN-based GoogleNet method for the extraction of feature vectors. Next, DTODCNN-CC technique uses the DTO technique for hyperparameter selection of GoogleNet approach. For crop classification purposes, DTODCNN-CC system deploys an extreme learning machine (ELM) algorithm. Finally, a modified sine cosine algorithm (MSCA) is introduced for the optimum parameter tuning of ELM methodology that results in an upgraded classification solution. To validate the better crop classification solution of DTODCNN-CC methodology, a wide range of experimental analyses were applied. The practical advantage enables automated and precise mapping of agricultural landscapes, facilitating efficient crop management and resource allocation by providing accurate crop classification data, DTODCNN-CC enables real-time monitoring of crop growth, health, and yield, allowing farmers to detect potential issues and take timely corrective actions. The practical issues with image quality, noise, or inaccurate annotations in the training data can negatively impact the model's performance and potentially lead to biased classifications.

This article is organized as follows: The "Literature Review" section presents a review of related work in RSI classification using deep learning. "The Proposed Model" describes the proposed DTODCNN-CC approach in detail. The "Results and Discussion" section discusses the results and compares DTODCNN-CC with existing methods. "Conclusions" concludes the article and discusses future research directions.

## LITERATURE REVIEW

*Munaf & Oguz (2024)* projected a plant disease detection for significant yield losses. Traditional detection methods, while still widely used, are frequently laborious and prone to errors, highlighting the need for more efficient, scalable, and rapid solutions. The potential of Deep Learning (DL) models, particularly Convolutional Neural Networks (CNNs) and MobileNet architectures, for early and accurate identification of plant diseases. Achieved an even higher accuracy of 96%, indicating its potential for real-world deployment on mobile devices. *Chamundeeswari et al. (2022)* projected an optimum

DCNN-based crop classification model (ODCNN-CCM) utilizing multispectral RSIs. This introduced ODCNN-CCM method primarily utilizes an adaptive wiener filter-based image preprocessing method. Furthermore, the RetinaNet algorithm was implemented for executing the feature extraction method. Lastly, a dolphin swarm optimizer (DSO) with a deep SDAE (DSDAE) algorithm was employed for classifying types of crops. *Karthikeyan et al. (2023)* introduced a novel remora optimizer with a DL-driven crop classification and chlorophyll contents estimation (RODLD-C4E) approach that exploited multispectral RSIs. To achieve this, this developed RODLD-C4E algorithm primarily arises using an RO method with NASNetLarge framework to extract features. The deployment of the RO method permits to effectually the adoption of the hyperparameters of the NasNetLarge framework. Furthermore, the cascaded GRU (CGRU) technique was utilized for classifying types of crops. Eventually, the DBN technique could use for evaluating the chlorophyll contents present in the crop.

*Meng et al. (2021)* considered a technique of DL-based crop mapping applying single-shot hyperspectral satellite images, there are 3-CNN approaches, namely, 1DCNN, 2DCNN, and 3DCNN methods, which can be implemented for endwise crop mapping. Additionally, a multiple learning-based visualization methodology like t-distributed stochastic neighbour embedding (t-SNE) mainly presented for representing the discriminated proficiency of deep semantic features extracted employing several CNN methods. *Qiao et al. (2021)* suggested a new DL technique for predicting crop production such as SSTNN (Spatial-Spectral-Temporal Neural Network) that incorporates 3D-CNN as well as RNN for employing their complementarities. Especially, SSTNN integrates temporal dependency capturing and SS learning methods into a combined convolutional network for identifying the integrated SS-temporal models.

In *Chew et al. (2020)*, the authors employed RGB images composed of UAVs flown in Rwanda to improve a DL approach to identify types of crops, especially legumes, bananas, and maize that can be major strategic food crops in Rwandan cultivation. This method leverages the development of DCNNs and TL, exploiting the openly available ImageNet database and VGG16 framework for pretraining. The authors (*Gadiraju et al., 2020*) introduced a method multimodal DL method that collectively utilizes SS and phenological characteristics for recognizing the main types of crops.

*Li, Shen & Yang (2020)* suggested a novel technique that integrates LSTM, GAN, and CNN frameworks for classifying crops of soybeans and corn from time-series RSIs that GAN's discriminator has been employed as the outcome classifier. This technique is possible in the case that the training instances can be lesser, and it entirely obtains the benefit of phenology, spatial, and spectral features of the crop from satellite information. In *Wei et al. (2019)*, an approach of SS fusion that depends on CRF (SSF-CRF) for classifying the crops in UAV-borne hyperspectral RSIs is introduced. This suggested technique develops appropriate possible functions in a pairwise CRF algorithm, combining the spatial and spectral features to decrease spectral dissimilarity in similar fields and efficiently detect crops. *Kanna et al. (2023)* proposed sophisticated deep learning approaches for early disease prediction in cauliflower plants using VegNet image dataset. Combines DCNN, ELM, and DTO for robust disease classification. Future direction

towards requires significant computational resources, black-box model. *Dhaka et al. (2021)* using Deep convolutional neural networks (DCNNs) using Public datasets like PlantVillage, Leafsnap. Leverages DCNNs for feature extraction and classification of plant leaf diseases. Requires large datasets for training, computationally expensive. *Kundu et al. (2021)* proposed automatic and intelligent data collector and classifier. Requires specialized hardware and expertise for setup and maintenance. Pearl millet farmland data, including imagery and parametric data. Combines IoT technology with interpretable deep transfer learning for real-time disease prediction.

## THE PROPOSED MODEL

In this article, mainly focused on the development and project of DTODCNN-CC algorithm for crop detection by RSI Analysis. DCNN-based GoogleNet the pre-trained GoogleNet architecture, with high accuracy in image recognition tasks. GoogleNet for extracting features from the RSIs that are relevant to crop detection and classification. DTO-based Hyperparameter Tuning optimizes and improve its performance on the specific task of crop detection. ELM-based Detection for the final detection of crops in the RSIs. MSCA-based Parameter Selection technique is used to select the most relevant part of the image for crop detection and improve detection accuracy. Figure 1 exemplifies overview of DTODCNN-CC technology.

### Optimal GoogleNet-based feature extraction

To extract an optimum set of features, GoogleNet model is applied. DL algorithm learns directly from images such as lower level, middle-level, and abstract features that are besides hand-crafted features (*Ashraf et al., 2020*). The trained group of images is applied as input for the GoogleNet pretrained model for extracting the deep features. In such cases, 144 layers are used together with FC layer, convolution layer, and groups of images by dissimilar modalities. It repositioned the features of GoogleNet model by using validation and training on medical image data. The objective is to carry out relearning based on 12 class labels of data.

GoogleNet has trained already for 1,000 semantic class labels by applying features in large-scale data. TL of deep network whereby it regenerates a network based on a new level by finetuning of parameters. During the finetuning model, the feature was extracted from the provided group of images. The max-pooling layer has been used for reducing the dimensional of input through the same padding, eight filters, kernels of $4 \times 4$, and a stride of 1. The outcome of pooling layers provides a $112 \times 112$-dimension outcome. The outcome of the initial convolution layer is given to non-linearity, subsequently, the spatial max-pooling layer summarizes neighboring neurons. ReLU can be applied for non-linearity to outcomes of the FC layer. For the pretraining GoogleNet model, images can be applied as input for replacing the final three layers, *viz.*, output, loss three-classifier, and prob, to rejoin the layer with residual network. In such cases, an FC layer, a softmax layer, and the classifier resultant layer are added to pretrained GoogleNet models. Softmax functions to N-dimension vector that measures the vector values within (0,1) and its

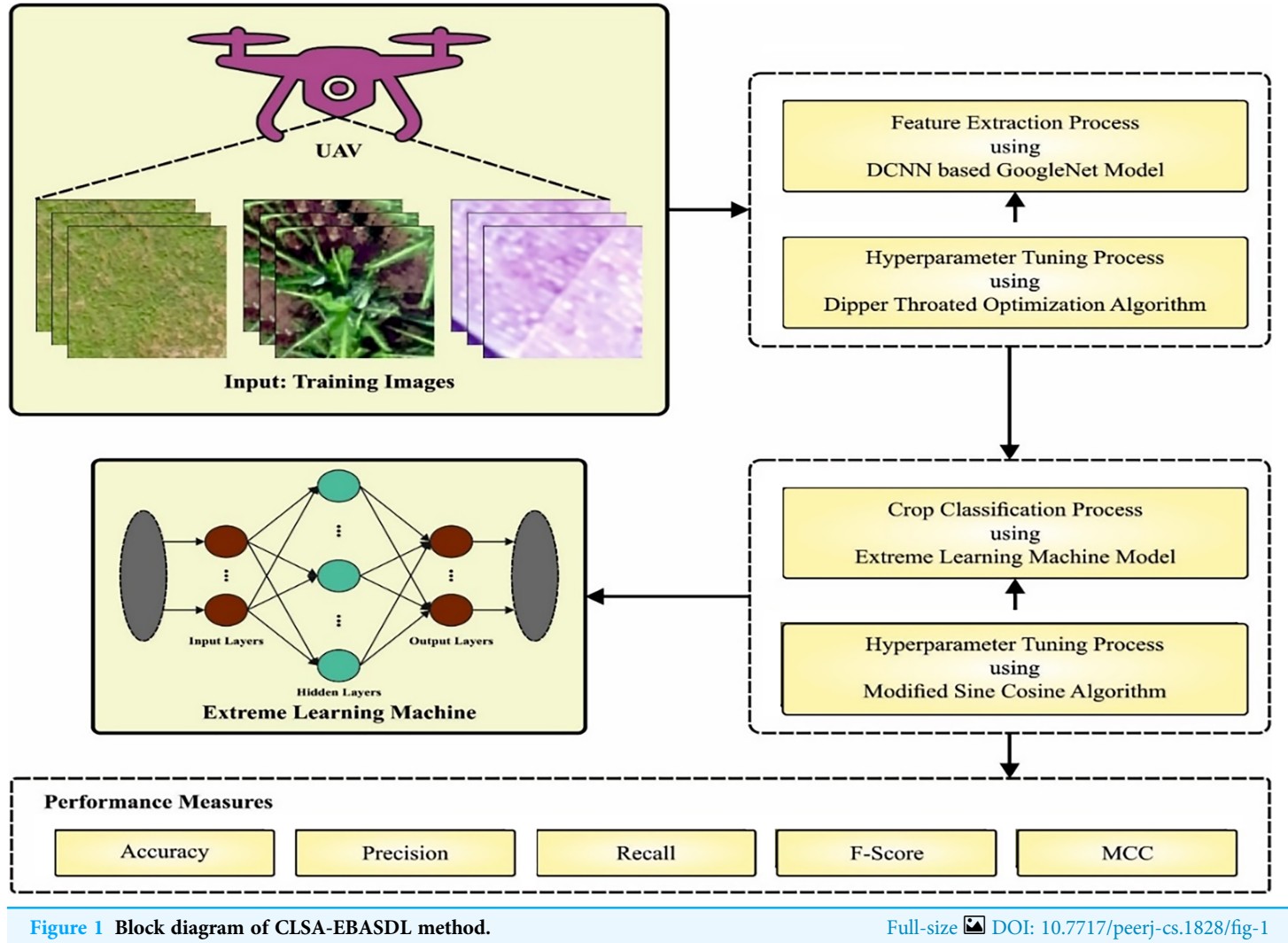

**Figure 1** Block diagram of CLSA-EBASDL method.     

summation provides a value of 1 to characterize all classes. The final FC layer is of a similar size *viz.*, the amount of classes of the data is 12.

For adjusting parameters connected to the GoogleNet, the DTO algorithm is introduced. DTO is a new metaheuristic technique enthused by the cooperative nature of Birds (*Abdelhamid et al., 2023*). A narrow mathematical model and a complete overview of its usage and discovery are discussed in this section. Three different techniques are used in the DTO technique to optimize exploration: (1) flying effectively over a known area, (2) flying towards a new site, and (3) shifting towards another bird.

A flock of bird's swims in space to find food while applying DTO algorithm. The speed and position of birds can be represented as P and V. DTO might explore the search range for the optimum solutions using these metrics.

$$P = \begin{bmatrix} P_{1,1} & P_{1,2} & P_{1,3} & \cdots & P_{1,d} \\ P_{2,1} & P_{2,2} & P_{2,3} & \cdots & P_{2,d} \\ P_{3,1} & P_{3,2} & P_{3,3} & \cdots & P_{3,d} \\ \cdots & \cdots & \cdots & \cdots & \cdots \\ P_{m,1} & P_{m,2} & P_{m,3} & \cdots & P_{m,d} \end{bmatrix} \tag{1}$$

$$V = \begin{bmatrix} V_{1,1} & V_{1,2} & V_{1,3} & \cdots & V_{1,d} \\ V_{2,1} & V_{2,2} & V_{2,3} & \cdots & V_{2,d} \\ V_{3,1} & V_{3,2} & V_{3,3} & \cdots & V_{3,d} \\ \cdots & \cdots & \cdots & \cdots & \cdots \\ V_{m,1} & V_{m,2} & V_{m,3} & \cdots & V_{m,d} \end{bmatrix} \tag{2}$$

For indexes $i \in 1, 2, 3, m$ and $j \in 1, 2, 3 \ldots$, in $j^{th}$ dimensional parameter, bird position $i^{th}$ represented as $P_{(i,j)}$, and bird speeds represented as $V_{(i,j)}$. The subsequent range defines the bird's fitness $f = f_1, f_2, f_3, \ldots, f_n$.

$$f = \begin{bmatrix} f_1\left(P_{1,1}, P_{1,2}, P_{1,3}, \cdots, P_{1,d}\right) \\ f_2\left(P_{2,1}, P_{2,2}, P_{2,3}, \ldots, P_{2,d}\right) \\ f_3\left(P_{3,1}, P_{3,2}, P_{3,3}, \cdots, P_{3,d}\right) \\ \cdots \\ f_m\left(P_{m,1}, P_{m,2}, P_{m,3}, \cdots, P_{m,d}\right) \end{bmatrix} \tag{3}$$

Mother birds basically contain maximal fitness between bird's since it is offered maximum offspring with the capability to search for survival and food. $P_{best}$ represents the best location in the search space. $P_{nd}$ signifies the regular bird, which serves as a follower of the mother bird. $P_{Gbest}$ indicates the global optimum solution.

$$X = P_{best}(i) - K_1.|K_2.P_{best}(i) - P(i)| \tag{4}$$

$$Y = P(i) + V(i+1) \tag{5}$$

$$P(i+1) = \begin{cases} X \ if \ r_3 < 05 \\ Y \ otherwise \end{cases} \tag{6}$$

$$V(i+1) = K_3 V(i) + K_4 r_1 (P_{best}(i) - P(i)) + K_5 r_2 (P_{Gbest} - P(i)) \tag{7}$$

Here $P_{best}(i)$ indicates the optimal position of the bird, $P(i)$ shows the average place of the bird at $i^{th}$ iteration, and $V(i+1)$ indicates bird speeds at $i+1$ iteration. $K_4$ and $K_5$ are said to be coefficients with values 1.7 and 1.8, correspondingly, then the weight values $K_1$, $K_2$, and $K_3$ dynamically nominated in the range [0–2]. $r_1, r_2,$ and $r_3$ are arbitrary number ranges from zero to one.

### Crop classification using ELM model

For crop classification system, ELM algorithm is used. The ELM model depends on the SLFN (*Huang et al., 2019*). The working principles and network structure of SLFN are discussed in the following.

Assume $(x_i, y_i), i = 1, 2, \ldots, N$, while $x_i = [x_{i1}, x_{i2}, \ldots x_{in}]^T \in R^n$, characterizes the sample features; $y_i = [y_{i1}, y_{i2}, \ldots, y_{im}]^T \in R^m$ signifies tags of $i^{th}$ samples in $m$ classes, and

$y_{ij} \in \{0,1\}, j = 1, 2, \ldots m$. SLEN architecture with activation function (x) and L hidden nodes ($L \leq N$). A SLFN is mathematically modelled as follows:

$$y_j = \sum_{i=1}^{L} \beta_i g(w_i, \ b_i, \ x_i), \ j = 1, 2, \ \ldots N \tag{8}$$

In Eq. (8), $\beta_i$ and $w_i$ show the input and output weights of $i^{th}$ nodes of the hidden layer (HL); $w_i$ and $b_i$ are random numbers, $b_i$ signifies the biases of $i^{th}$ node of HL; $g(w_i, b_i, x_i)$ indicates activation function of $i^{th}$ nodes in HL. Equation (8) is expressed as follows:

$$H\beta = Y \tag{9}$$

whereas,

$$
\begin{aligned}
H &= H(w1, \ \ldots, \ w_L, \ b_1, \ \ldots, \ b_L, x_1, \ \ldots, \ x_N) \\
&= \begin{bmatrix} g(w_1, b_1, x_1) & \cdots & g(w_L, b_L, x_1) \\ \vdots & \cdots & \vdots \\ g(w_1, b_1, x_N) & \cdots & g(w_L, b_L, x_N) \end{bmatrix}_{N \times L} \\
\beta &= \begin{bmatrix} \beta_1^T \\ \vdots \\ \beta_L^T \end{bmatrix}, \ Y = \begin{bmatrix} y_1^T \\ \vdots \\ y_N^T \end{bmatrix}
\end{aligned}
$$

The parameter of SLFN is evaluated by the minimum-squares solution:

$$\min ||H\beta - Y|| \tag{10}$$

On the other hand, ELM finds a series of optimum parameters $\hat{\beta}, W_i, \hat{b}, i = 1, 2, \ldots, L$, thus:

$$||H\left(\hat{w}_1, \ \ldots, \hat{w}_L, \hat{b}_1, \ \ldots, \hat{b}_L\right)\hat{\beta} - Y|| = \min_{\beta, w_i, b_i} ||H\left(\hat{w}_1, \ \ldots, \hat{w}_L, \hat{b}_1, \ \ldots, \hat{b}_L\right)\beta - Y|| \tag{11}$$

The least-square solution of this formula is:

$$\hat{\beta} = H^+ Y \tag{12}$$

In Eq. (12), $H^+$ refers to the MP generalized inverse of the matrix H.

## Optimal hyperparameter turning using MSCA

Finally, MSCA was performed parameter tuning of ELM system. The optimization process of SCA begins with a set of arbitrary solutions as a starting point (*Mani, Shaker & Jovanovic, 2023*). Next, this solution is enhanced by a set of procedures that form the basis of the optimizer algorithm. The efficiency can be measured by the objective function. These two phases of optimization techniques such as exploration and exploitation, are of equal importance.

In the exploration phase, the optimization technique fuses a random solutions with a high degree of unpredictability to identify the possible area within the searching range

(*Dadi, Tamilvizhi & Surendran, 2022*; *Singh Gill et al., 2022*). But as the process transitions to the exploitation stage, the random solution undergoes progressive modification, and the level of random variation reduces considerably than the exploration stage. For updating the position, a subsequent equation is developed for these two stages:

$$X_i^{t+1} = X_i^t + r1 \times \sin(r2) \times \left| r3P_i^t - X_i^t \right| \tag{13}$$

$$X_i^{t+1} = X_i^t + r1 \times \cos(r2) \times \left| r3P_i^t - X_{ii}^t \right| \tag{14}$$

where $X_i^t$ is the location of the existing answer in [^] $i^{th}$ parameter at $t^{th}$ iteration. r1, r2, and r3 are arbitrary values, and $P_i$ shows the place of the target fact in $i^{th}$ dimension. The complete value is denoted by $\|$. The combination of these two equations is given below:

$$X_i^{t+1} = \begin{cases} X_i^t + r1 \times \sin(r2) \times \left| r3P_i^t - X_i^t \right|, & r4 < 0.5 \\ X_i^t + r1 \times \cos(r2) \times \left| r3P_i^t - X_{ii}^t \right|, & r4 \geq 0.5 \end{cases} \tag{15}$$

In Eq. (15), r4 denotes the random integer within [0,1].

The SCA technique integrates four essential parameters: r1, r2, r3, and r4. During the optimization process, the parameter r1 defines the location area. A parameter r2 defines the magnitude of movement. The parameter r3 controls the effect of the endpoint on the solution. Finally, the parameter r4 switches between sin and cos functions in Eq. (15). In certain executions, SCA demonstrates poorer performance caused by an extreme focus on the less promising regions of the searching region. This leads to a total reduced quality of outcomes.

Fortunately, algorithm hybridization is a conventional method to overcome known deficiencies of optimization algorithms (*Gill et al., 2022*; *Surendran, Alotaibi & Subahi, 2023a*). This integrates the famous FA search model into the basic SCA algorithm to overcome the lack of exploratory ability.

$$X_i^{(t+1)} = X_i^t + \beta_0 \cdot e^{\left( -\gamma r_{(i,j)}^2 \right)} \left( X_j^t - X_i^t \right) + \alpha^t \left( \kappa - 0.5 \right) \tag{16}$$

In Eq. (16), α is the random parameter, κ shows the pseudo-random number derived from the Gaussian distribution. The range amongst i and j individuals is represented as $r_{(i,j)}$. γ parameter defines the light propagating characteristics of the media and the solution quality $\beta_0$ defines the outcomes of the objective function. Figure 2 depicts the stages involved in MSCA.

The resultant low-level hybrid algorithm incorporates one further mechanism to maintain better stability. The φ parameter is attached to all the solutions. In all iterations, the pseudo-random values range from zero to one. When the φ value is higher than 0.5, then the FA search model is used. Or else, the typical SCA search is used. This model is only enabled following 1/3 of the initial iterations to encourage stability.

The MSCA system developments an FF to make the best detection algorithm answer. It describes an optimistic number to signify the optimum results of candidate efficiency. During this situation, the decrease in classifier errors assumed that FF (*Rineer et al., 2021*).

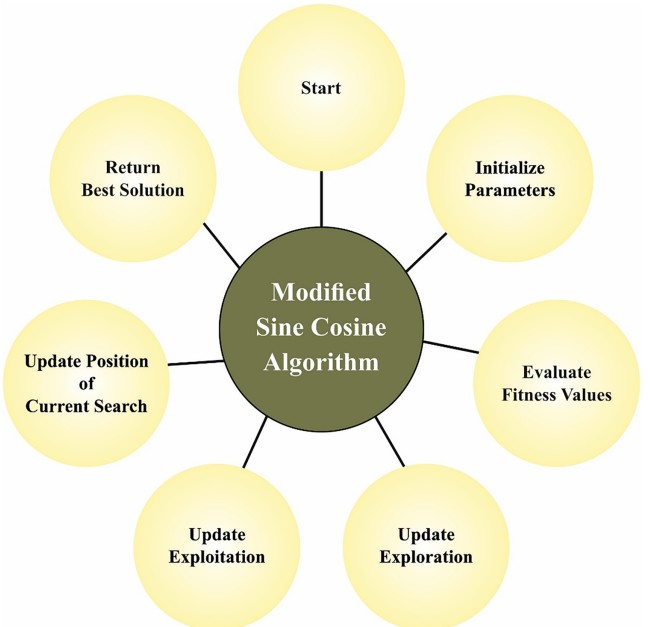

**Figure 2 Steps involved in MSCA.**

The parameters of the GoogleNet architecture algorithm has Number of layers and their configurations (*e.g.*, filter sizes, activation functions), and Pre-trained weights used for transfer learning. Dipper Throated Optimization (DTO) algorithm optimization parameters (*e.g.*, Levy flight coefficient, search space boundaries). Extreme learning machine (ELM) algorithm regularization parameter. Modified sine cosine algorithm (MSCA) algorithm optimize size and termination criteria.

$$fitness(x_i) = ClassifierErrorRate(x_i) = \frac{No.\ of\ misclassified\ instances}{Total\ no.\ of\ instances} * 100 \qquad (17)$$

## RESULTS AND DISCUSSION

The simulation validation of DTODCNN-CC technology was verified on the drone imagery dataset (*Surendran, Alotaibi & Subahi, 2023b*), including 6,450 examples with six classes determined in Table 1. The size of the dataset (6,450 images) is relatively large for crop classification tasks. This provides a good foundation for training deep learning models like DTODCNN-CC. However, the number of images for certain classes might be insufficient for achieving optimal performance, particularly for Legume and Structure.

Figure 3 establishes the confusion matrices created by DTODCNN-CC technique under 80:20 and 70:30 of the TRS/TSS. The result value denotes effectual detection and detection of all six classes.

The average MCC values for both the training and test sets are above 98%, indicating excellent performance for the overall classification task. The average precision and recall values are also high, ranging from 92.99% to 93.12% for precision and from 91.00% to 91.44% for recall. The F1-score, which combines precision and recall, shows similar

**Table 1 Detailed database.**

| Classes | No. of images |
|---|---|
| Maize | 2,075 |
| Banana | 1,661 |
| Forest | 1,270 |
| Other | 750 |
| Legume | 363 |
| Structure | 331 |
| Total no. of images | 6,450 |

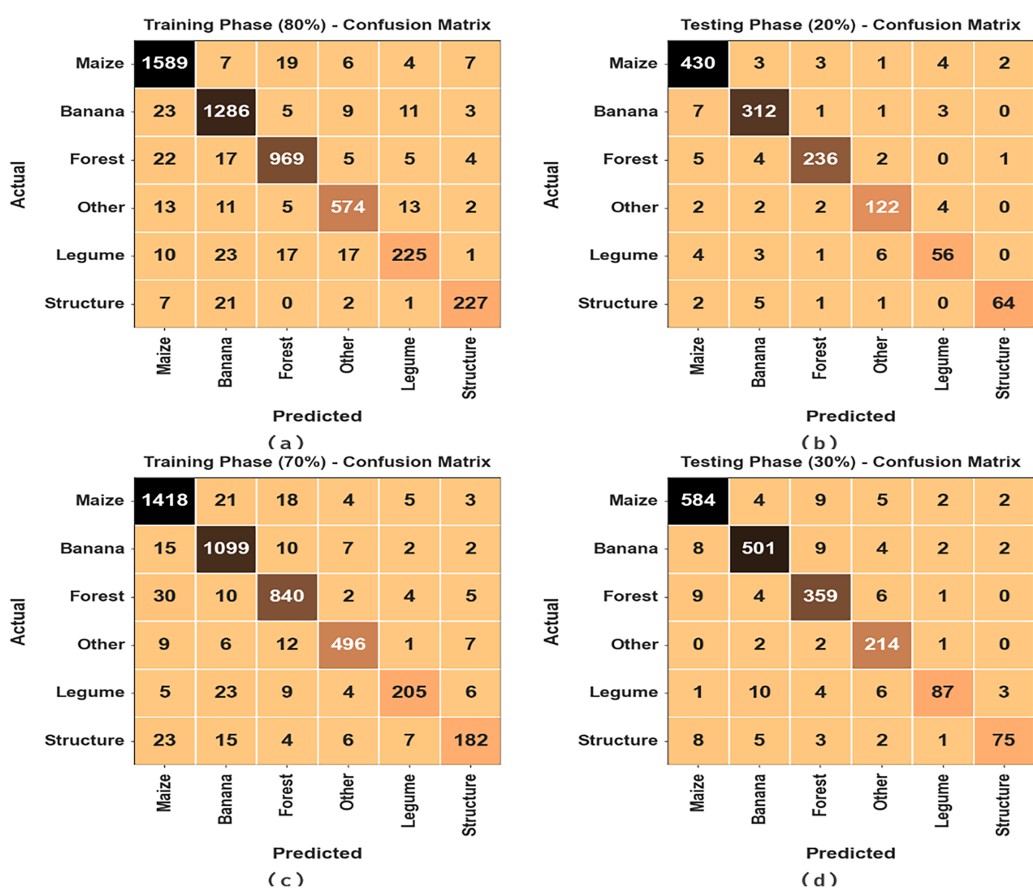

**Figure 3 Confusion matrices of (A and B) 80:20 of TR set/TS set and (C and D) 70:30 of TR set/TS set.**

performance, with an average value of 91.99% on the training set and 92.18% on the test set. The "Structure" class exhibits the highest performance across all metrics, achieving an average MCC of 99.07%, precision of 94.75%, recall of 90.44%, and F1-score of 91.24%. The "Legume" class shows the lowest performance, particularly in terms of recall and F1-score, indicating that the model struggles to correctly identify all instances of this class. The crop classification results of DTODCNN-CC model at 80:20 of TRS/TSS are portrayed in

**Table 2 Crop classifier outcome of DTODCNN-CC algorithm at 80:20 of TR set/TS set.**

| Classes | $Accu_y$ | $Prec_n$ | $Reca_l$ | $F_{Score}$ | MCC |
|---|---|---|---|---|---|
| **TR set (80%)** | | | | | |
| Maize | 97.71 | 95.49 | 97.37 | 96.42 | 94.75 |
| Banana | 97.48 | 94.21 | 96.19 | 95.19 | 93.49 |
| Forest | 98.08 | 95.47 | 94.81 | 95.14 | 93.95 |
| Other | 98.39 | 93.64 | 92.88 | 93.26 | 92.35 |
| Legume | 98.02 | 86.87 | 76.79 | 81.52 | 80.65 |
| Structure | 99.07 | 93.03 | 87.98 | 90.44 | 89.99 |
| **Average** | **98.13** | **93.12** | **91.00** | **91.99** | **90.86** |
| **TS set (20%)** | | | | | |
| Maize | 97.44 | 95.56 | 97.07 | 96.30 | 94.36 |
| Banana | 97.75 | 94.83 | 96.30 | 95.56 | 94.06 |
| Forest | 98.45 | 96.72 | 95.16 | 95.93 | 94.98 |
| Other | 98.37 | 91.73 | 92.42 | 92.08 | 91.17 |
| Legume | 98.06 | 83.58 | 80.00 | 81.75 | 80.75 |
| Structure | 99.07 | 95.52 | 87.67 | 91.43 | 91.03 |
| **Average** | **98.19** | **92.99** | **91.44** | **92.18** | **91.06** |

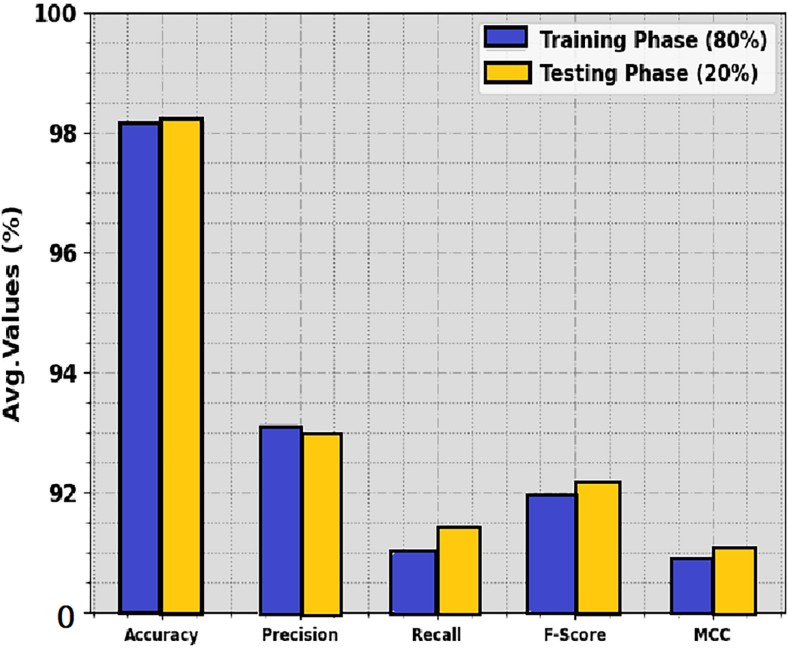

**Figure 4 Average of DTODCNN-CC algorithm on 80:20 of TR set/TS set.**

Table 2 and Fig. 4. The results show that DTODCNN-CC method suitably recognizes six class labels. With an 80% TRS, DTODCNN-CC technology offers average $accu_y$, $prec_n$, $reca_l$, $F_{score}$, and MCC of 98.13%, 93.12%, 91%, 91.99%, and 90.86%, respectively.

**Table 3 Crop classifier outcome of DTODCNN-CC algorithm at 70:30 of TR set/TS set.**

| Classes | $Accu_y$ | $Prec_n$ | $Reca_l$ | $F_{Score}$ | MCC |
|---|---|---|---|---|---|
| **TR set (70%)** | | | | | |
| Maize | 97.05 | 94.53 | 96.53 | 95.52 | 93.34 |
| Banana | 97.54 | 93.61 | 96.83 | 95.19 | 93.57 |
| Forest | 97.70 | 94.06 | 94.28 | 94.17 | 92.74 |
| Other | 98.72 | 95.57 | 93.41 | 94.48 | 93.76 |
| Legume | 98.54 | 91.52 | 81.35 | 86.13 | 85.53 |
| Structure | 98.27 | 88.78 | 76.79 | 82.35 | 81.69 |
| **Average** | **97.97** | **93.01** | **89.86** | **91.31** | **90.10** |
| **TS set (30%)** | | | | | |
| Maize | 97.52 | 95.74 | 96.37 | 96.05 | 94.25 |
| Banana | 97.42 | 95.25 | 95.25 | 95.25 | 93.47 |
| Forest | 97.57 | 93.01 | 94.72 | 93.86 | 92.35 |
| Other | 98.55 | 90.30 | 97.72 | 93.86 | 93.13 |
| Legume | 98.40 | 92.55 | 78.38 | 84.88 | 84.36 |
| Structure | 98.66 | 91.46 | 79.79 | 85.23 | 84.74 |
| **Average** | **98.02** | **93.05** | **90.37** | **91.52** | **90.38** |

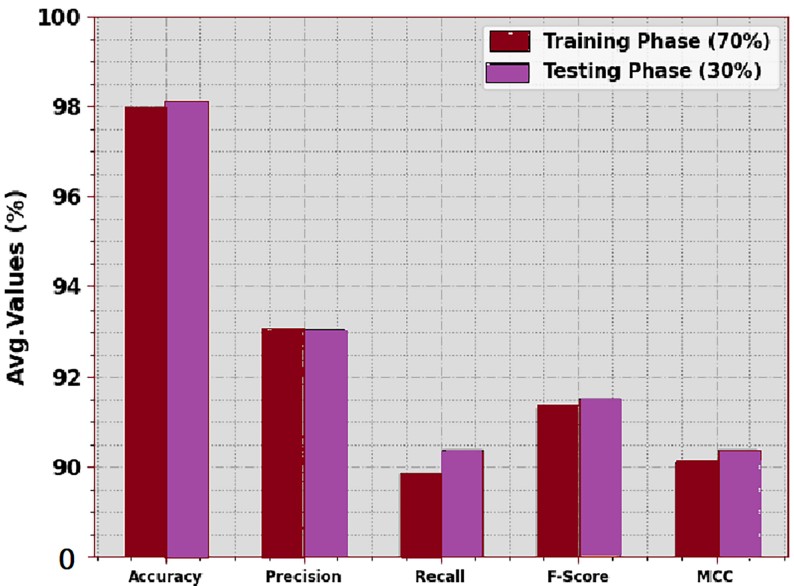

**Figure 5 Average of DTODCNN-CC algorithm on 70:30 of TR set/TS set.**

Moreover, on the 20% TSS, DTODCNN-CC methodology achieves average $accu_y$, $prec_n$, $reca_l$, $F_{score}$, and MCC of 98.19%, 92.99%, 91.44%, 92.18%, and 91.06% correspondingly.

The crop classifier outcome of DTODCNN-CC methodology at 70:30 of TRS/TSS is described in Table 3 and Fig. 5. The simulation value denotes that DTODCNN-CC model properly identifies six classes. With 70% TRS, DTODCNN-CC algorithm achieves average

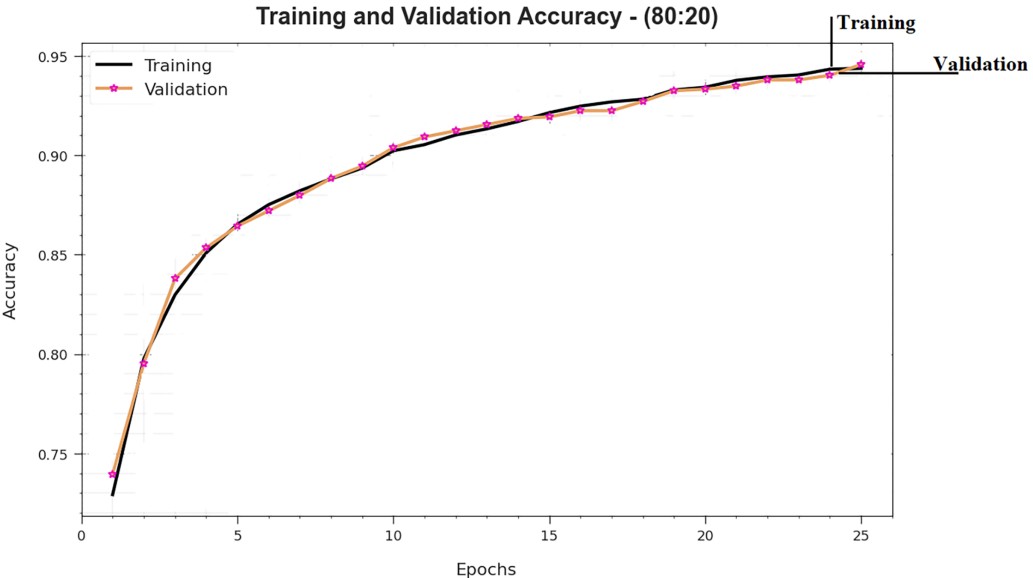

**Figure 6 Accuracy curve of DTODCNN-CC algorithm on 80:20 of TR set/TS set.**

$accu_y$, $prec_n$, $reca_l$, $F_{score}$, and MCC of 97.97%, 93.01%, 89.86%, 91.31%, and 90.10% correspondingly. Finally, with 30% TSS, DTODCNN-CC method attains average $accu_y$, $prec_n$, $reca_l$, $F_{score}$, and MCC of 98.02%, 93.05%, 90.37%, 91.52%, and 90.38% similarly.

To estimate the performance of DTODCNN-CC technology with 80:20 of TRS/TSS, TRS, and TSS $accu_y$ curves distinct those exposed in Fig. 6. TRS and TSS $accu_y$ curves create DTODCNN-CC performance technique over numerous epochs. The figure provides expressive particulars about learning, challenge, and simplification capacities of DTODCNN-CC technique. With a growth in epoch count, it is observed that TRS and TSS $accu_y$ curves get upgraded. It is perceived that DTODCNN-CC technique attains greater testing accurateness that can recognize designs in TRS and TSS data.

Figure 7 shows the complete TRS and TSS loss values of DTODCNN-CC model with 80:20 of TRS/TSS over epochs. TRS loss display the model loss gets diminished above epochs. Chiefly, the loss values obtained decreased as the model alters the load to decrease the forecast fault on TRS and TSS data. The loss curves determine the range that the method fits the training data. It is noticed that TRS and TSS loss progressively reduced as well as depicted that DTODCNN-CC model successfully learns the patterns shown in TRS and TSS data. As well, it is seen that DTODCNN-CC model adjusts the parameters to diminish the difference amongst the prediction and original training labels.

The precision-recall curve of DTODCNN-CC model with 80:20 of TRS/TSS is established by plotting the accuracy beside recall as clear in Fig. 8. The outcomes confirm that DTODCNN-CC technique gets amplified precision recall values below all classes. The figure portrays that the model learns to identify numerous class labels. DTODCNN-CC technique completes the enhanced outcomes in the detection of positive samples with minimum false positives (*Gao et al., 2021*).

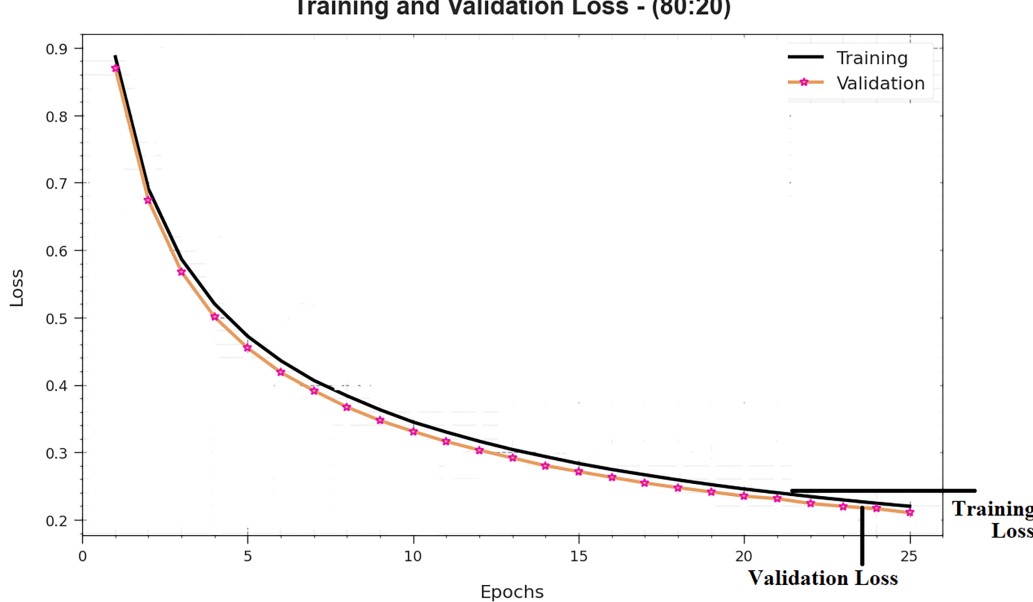

**Figure 7  Loss curve of DTODCNN-CC algorithm on 80:20 of TR set/TS set.**

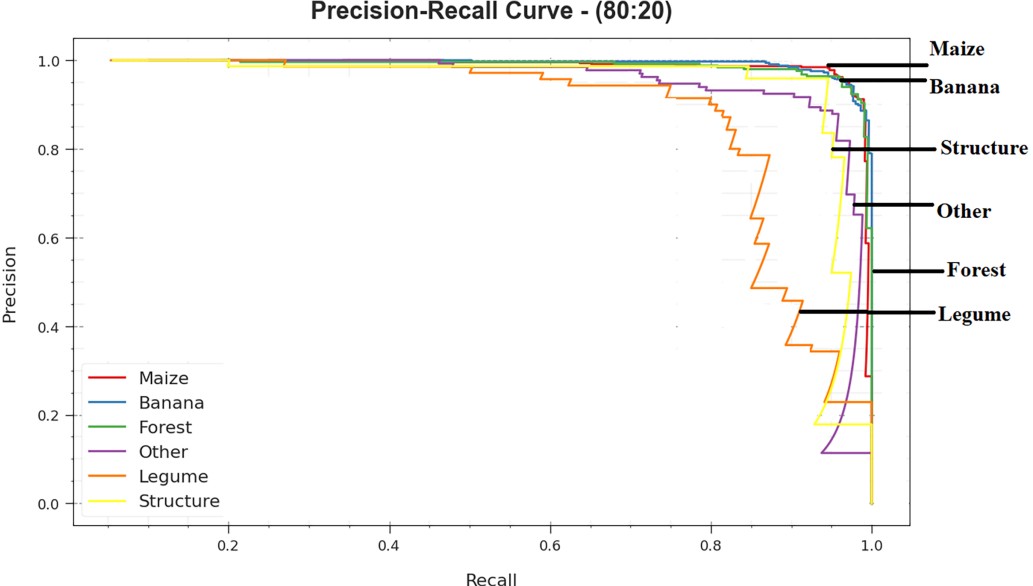

**Figure 8  PR curve of DTODCNN-CC algorithm on 80:20 of TR set/TS set.**

The ROC curves provided by DTODCNN-CC technique with 80:20 of TRS/TSS are demonstrated in Fig. 9, ability to differentiate class labels. The figure indicates respected vision of trade-off amongst TPR and FPR rates above separate detection thresholds as well as changing numbers of epochs. It projects the precise analytical performance of DTODCNN-CC methodology on the detection of distinct classes.

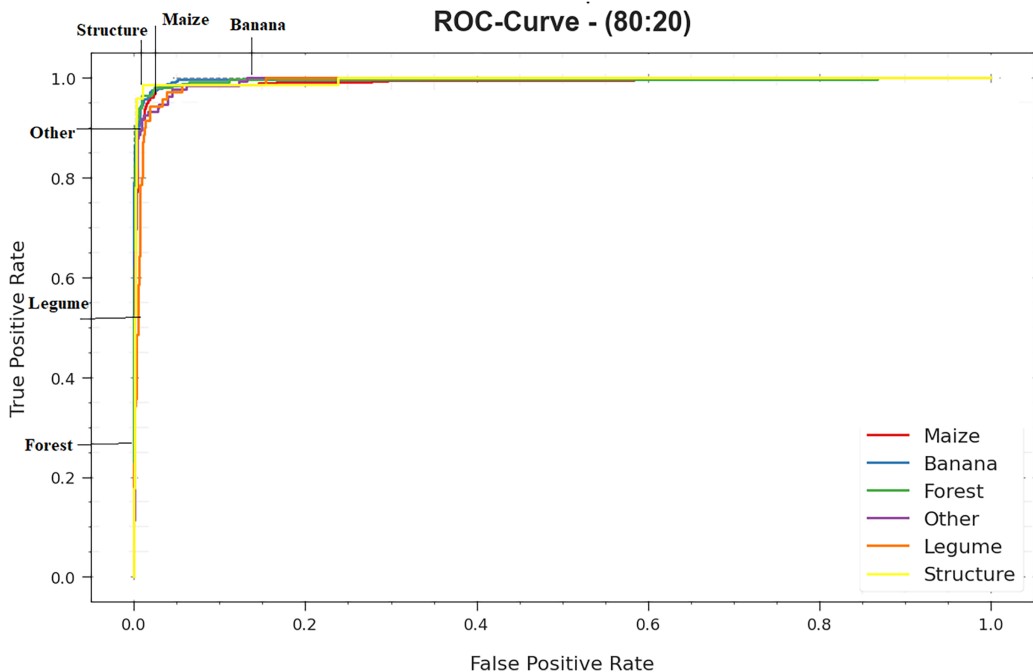

**Figure 9 ROC curve of DTODCNN-CC algorithm on 80:20 of TR set/TS set.**

**Table 4 Comparative outcome of DTODCNN-CC method with other systems.**

| Methods | $Accu_y$ | $Prec_n$ | $Reca_l$ | $F_{Score}$ |
|---|---|---|---|---|
| DTODCNN-CC | 98.19 | 92.99 | 91.44 | 92.18 |
| SBODL-FCC | 97.91 | 89.55 | 85.51 | 87.16 |
| DNN model | 86.69 | 86.66 | 84.85 | 86.74 |
| Alex Net | 91.00 | 88.15 | 82.24 | 83.81 |
| VGG-16 Model | 90.86 | 85.73 | 81.79 | 86.23 |
| ResNet model | 88.21 | 86.88 | 81.64 | 83.48 |
| SVM model | 87.17 | 88.49 | 84.15 | 84.75 |

In Table 4, the overall comparative results of DTODCNN-CC method are demonstrated (*Rineer et al., 2021*; *Surendran, Alotaibi & Subahi, 2023b*). Figure 10 represents the comparison analysis of the DTODCNN-CC method in terms of $accu_y$. The result stated enhancement of DTODCNN-CC technology in terms of $accu_y$. Depend on $accu_y$, DTODCNN-CC method acquires an aggregate $accu_y$ of 98.19%, whereas SBODL-FCC, DNN, AlexNet, VGG-16, ResNet, and SVM approaches get reduced $accu_y$ of 97.91%, 86.69%, 91%, 90.86%, 88.21%, and 87.17% correspondingly.

Figure 11 signifies the comparison analysis of DTODCNN-CC methodology in terms of $prec_n$, $reca_l$, and $F_{score}$. The simulation values inferred the developments of DTODCNN-CC method in terms of $prec_n$, $reca_l$, and $F_{score}$. Depend on $prec_n$, DTODCNN-CC system gains enhanced $prec_n$ of 92.99%, whereas SBODL-FCC, DNN, AlexNet, VGG-16, ResNet,

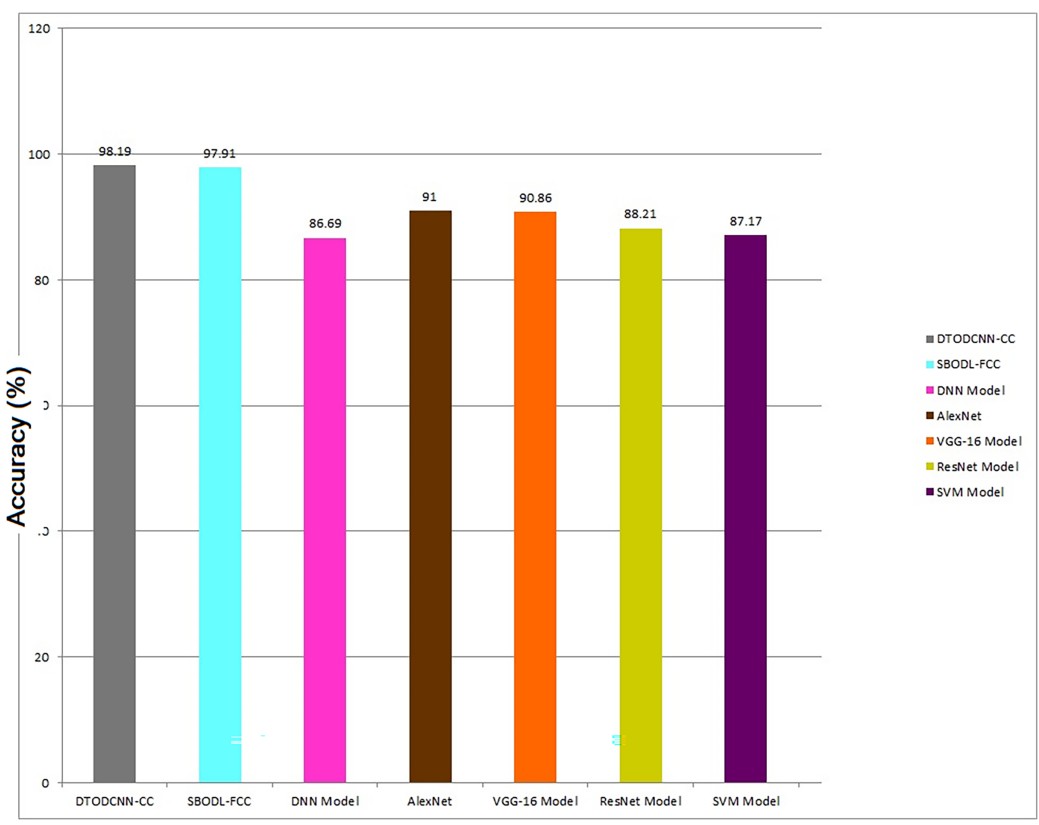

**Figure 10 Comparative outcome of DTODCNN-CC methodology with recent methods.**

and SVM algorithms attain lesser $prec_n$ of 89.55%, 86.66%, 88.15%, 85.73%, 86.88%, and 88.49% correspondingly. Moreover, with respect to $reca_l$, the DTODCNN-CC system achieves a higher $reca_l$ of 91.44%, whereas the SBODL-FCC, DNN, AlexNet, VGG-16, ResNet, and SVM models obtain lower $reca_l$ of 85.51%, 84.85%, 82.24%, 81.79%, 81.64%, and 84.15% correspondingly. Finally, based on $F_{score}$, DTODCNN-CC methodology gains an increasing $F_{score}$ of 92.18%, whereas SBODL-FCC, DNN, AlexNet, VGG-16, ResNet, and SVM systems obtain minimal $F_{score}$ of 87.16%, 86.74%, 83.81%, 86.23%, 83.48%, and 84.75% respectively. Thus, DTODCNN-CC model was utilized for enhanced crop classification results.

The dataset might not be representative of all potential scenarios and geographical regions. This can limit the generalizability of the model's performance to diverse environments and agricultural practices. Choosing optimal hyperparameters for the various algorithms within DTODCNN-CC can be challenging and time-consuming. This can lead to suboptimal performance if not carefully tuned. Combines the strengths of deep learning (DCNN), extreme learning machines (ELM), and metaheuristics (MSCA) to achieve efficient and accurate crop classification.

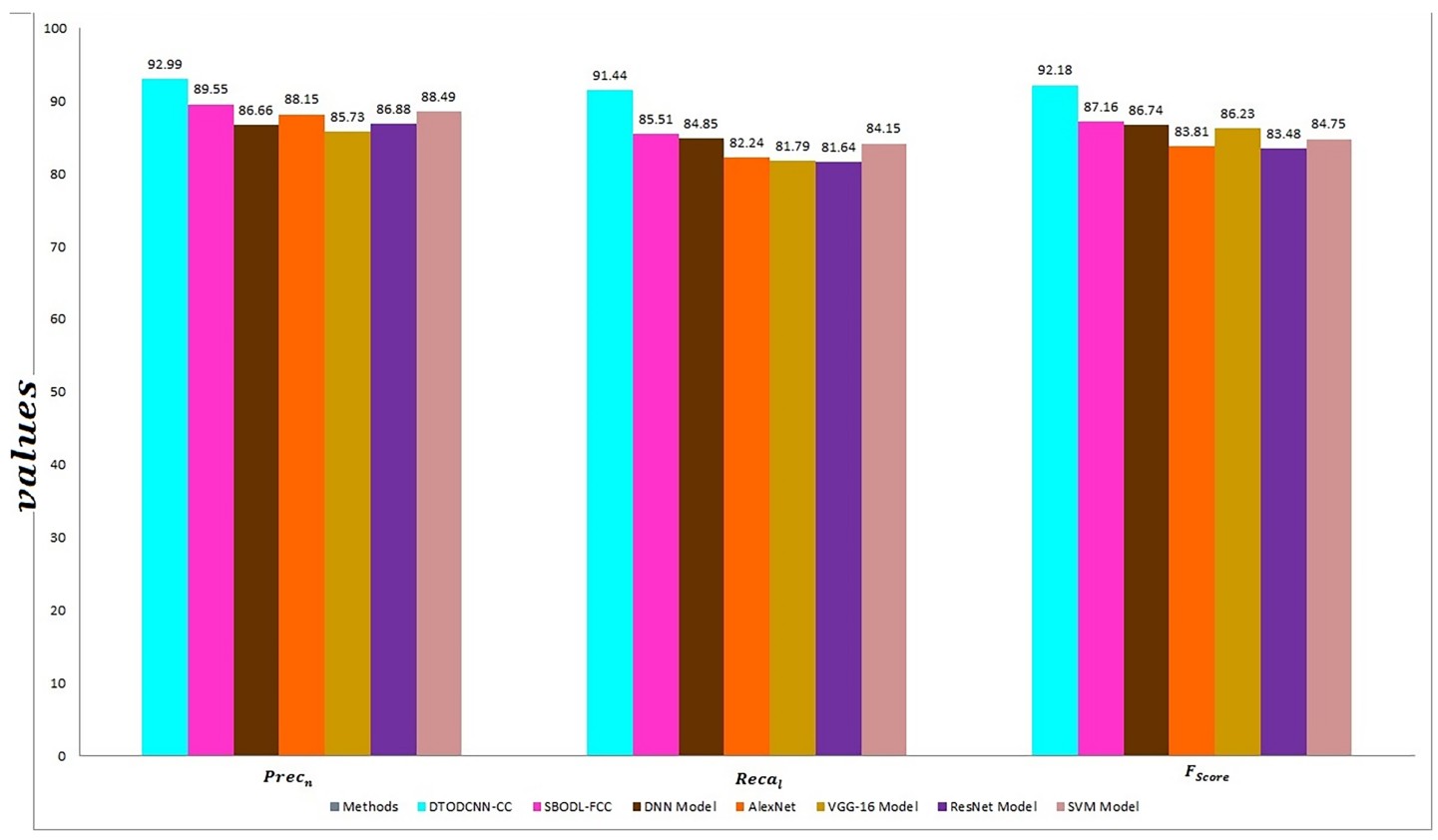

**Figure 11 Prec_n, reca_l, and F_score outcome of DTODCNN-CC approach with recent methods.**

## CONCLUSION

In this manuscript, we mainly concentrated on the growth and project of DTODCNN-CC algorithm for crop classification by RSI Analysis. The DTODCNN-CC architecture combines deep learning (DL) with extreme learning machines (ELM) and metaheuristics, providing a theoretically sound framework for efficient and accurate crop classification. The employment of the modified sine cosine algorithm (MSCA) for optimizing ELM parameters introduces a novel theoretical contribution to machine learning and optimization research. The DTODCNN-CC algorithm demonstrates the potential of integrating different machine learning techniques for achieving superior performance in complex classification tasks. The presented DTODCNN-CC algorithm concentrates to the detection and classification of food crops that exist in RSIs. To accomplish this, DTODCNN-CC technique applies DCNN-based GoogleNet model for the extraction of feature vectors. Next, DTODCNN-CC technology employs DTO model for hyperparameter selection of GoogleNet model. For crop classification purposes, DTODCNN-CC technique employs ELM model. Finally, MSCA is introduced for the optimum parameter tuning of ELM approach outcomes in an amended classification solution. To validate a better crop classification solution of DTODCNN-CC methodology, a large array of experimental evaluates are executed. DTODCNN-CC system gains enhanced prec_n of 92.99%, whereas SBODL-FCC, DNN, AlexNet, VGG-16, ResNet, and

SVM algorithms attain lesser prec$_n$ of 89.55%, 86.66%, 88.15%, 85.73%, 86.88%, and 88.49% correspondingly. An extensive result highlight the greater solution of DTODCNN-CC technique to other DL approaches in terms of different evaluation metrics.

### Funding

The current research study has received financial support from the Deanship for Research & Innovation, Ministry of Education in Saudi Arabia, under the auspices of project number: IFP22UQU4281768DSR120. The funders had no role in study design, data collection and analysis, decision to publish, or preparation of the manuscript.

### Grant Disclosures

The following grant information was disclosed by the authors:
Deanship for Research & Innovation, Ministry of Education in Saudi Arabia: IFP22UQU4281768DSR120.

### Competing Interests

The authors declare that they have no competing interests.

### Author Contributions

- Youseef Alotaibi conceived and designed the experiments, authored or reviewed drafts of the article, code, and approved the final draft.
- Brindha Rajendran conceived and designed the experiments, analyzed the data, performed the computation work, prepared figures and/or tables, and approved the final draft.
- Geetha Rani K. performed the experiments, analyzed the data, performed the computation work, prepared figures and/or tables, and approved the final draft.
- Surendran Rajendran performed the experiments, performed the computation work, prepared figures and/or tables, authored or reviewed drafts of the article, data set, and approved the final draft.

### Data Availability

The data is available at Zenodo: Surendran, R. (2023). multi class dataset_Dipper Throated Optimization with Deep Convolutional Neural Network-based Crop Classification on Remote Sensing Image Analysis [Data set]. Zenodo. https://doi.org/10.5281/zenodo.10361051.

The code is available at Zenodo: surendran, R. (2023). Dipper Throated Optimization with Deep Convolutional Neural Network-based Crop Classification on Remote Sensing Image Analysis. Zenodo. https://doi.org/10.5281/zenodo.10369740.

## Supplemental Information

Supplemental information for this article can be found online at http://dx.doi.org/10.7717/peerj-cs.1828#supplemental-information.

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
