# Peer review of "Dipper throated optimization with deep convolutional neural network-based crop classification for remote sensing image analysis"

_PeerJ Computer Science, doi:10.7717/peerj-cs.1828_

## Round 0.1 · original submission · Major Revisions

Reviewers suggested important changes to the paper, consider them in your revised work.

Reviewer 1 ·

Basic reporting

1. The manuscript fails to present clear and unambiguous, professional English.
2. The literature review is not comprehensive.
3. The results are not supported with the analysis.
4. The technical part contains many errors.

Experimental design

1. It is really confusing about the problem definition.
2. It is not clear about the contributions of this paper.
3. The proposed method is not explained with details.

Validity of the findings

1. The authors are trying to fool the editor and reviewers with some irrelevant code. This behavior should not be encouraged.
2. No data is shared.

Reviewer 2 ·

Basic reporting

The terminology surrounding remote sensing and deep learning can be quite advanced. It may be worth a deeper explanation of certain issues, e.g. algorithms related to the proposed method.
The article contains a literature review, but it is not very rich. I recommend authors to read, among others: work: Advanced deep learning techniques for early disease prediction in cauliflower plants; A survey of deep convolutional neural networks applied for prediction of plant leaf diseases; IoT and interpretable machine learning based framework for disease prediction in pearl millet.
The mathematical text in the article needs improvement - if variables or formulas appear in the text, they must also be written in a mathematical setting (see lines 187-189 and further in many places). Highlighted patterns are better, but they could be even better - e.g. pattern 16 is poorly formatted (I recommend the LaTeX editor). You should also remember about punctuation - also in mathematical formulas. The numbers of highlighted patterns should be aligned to the right - there is a bit of chaos in the article on this topic.

Experimental design

The introduction is clear and understandable, the research goal is clearly formulated, and the DTODCNN-CC algorithm proposal is clearly presented. The authors use appropriate references to models, such as GoogleNet, which increases the quality of the work.

Validity of the findings

The conclusion effectively summarizes the main aspects of the paper, highlighting the effectiveness of the proposed algorithm based on the experimental evaluation results. But it would be good to repeat at least some of the results "quantitatively" in this summary, as they emphasize the quality of the proposed method.
Can the authors also provide any contraindications or limitations of the proposed algorithm?

Additional comments

The article is interesting and raises important issues, but its major drawback is the editing page, which needs to be thoroughly improved.

Reviewer 3 ·

Basic reporting

See below

Experimental design

See below

Validity of the findings

See below

Additional comments

>> The language usage throughout this paper need to be improved, the author should do some proofreading on it.
>> Your abstract does not highlight the specifics of your research or findings. Rewrite the Abstract section to be more meaningful. I suggest to be Problem, Aim, Methods, Results, and Conclusion.
>> Introduction section can be extended to add the issues in the context of the existing work and how proposed algorithms/approach can be used to overcome this.
>> The problems of this work are not clearly stated. There is ambiguity in statement understanding.
>> Add main contributions list as points in the Introduction section.
>> Add the rest organization section at the end of the Introduction section.
>> More clarifications and highlighted about the research gabs in the related works section. I suggest to disccuss other studies, for example
- Deep Learning for Plant Disease Detection. International Journal of Mathematics, Statistics, and Computer Science, 2, 75–84. https://doi.org/10.59543/ijmscs.v2i.8343
>> I feel that more explanation would be need on how the proposed method is performed.
>> How does the progress in remote sensing (RS) technology contribute to the improvement of RS data quality, and in what fields is multi-source RS data extensively utilized?
>> What is the significance of remote sensing image (RSI) classification in various fields, and how does it support the creation of land use or land cover maps using RS data?
>> How has deep learning (DL), particularly Convolutional Neural Network (CNN), proven to be effective in computer vision (CV) tasks and what development opportunities does it offer for intelligent information extraction from RSIs?
>> Can you elaborate on the specific challenges or requirements in crop classification from RSIs that the proposed Dipper Throated Optimization with Deep Convolutional Neural Network-based Crop Classification (DTODCNN-CC) aims to address?
>> What role does the GoogleNet model play in the DTODCNN-CC technique for the extraction of feature vectors, and why was it chosen for this particular study?
>> How does the Dipper Throated Optimization (DTO) algorithm contribute to hyperparameter selection for the GoogleNet model, and what advantages does it offer in the context of RS image analysis?
>> What is the rationale behind using the extreme learning machine (ELM) methodology for crop classification in the DTODCNN-CC technique, and how does it complement the deep convolutional neural network (DCNN) based features?
>> Can you explain the significance of the modified sine cosine algorithm (MSCA) in optimizing parameters for the ELM approach, and how does it contribute to improving the classification results?
>> How is the proposed DTODCNN-CC methodology validated, and what specific experimental analyses are conducted to assess its performance in crop classification?
>> What evaluation metrics are considered in comparing the performance of the DTODCNN-CC system with other DL systems, and how does it outperform them in terms of different metrics?
>> Are there any limitations or challenges encountered during the implementation of the DTODCNN-CC methodology, and how were they addressed in the study?
>> Considering the application of the DTODCNN-CC technique, how scalable and adaptable is the proposed approach to different geographical regions and environmental conditions in the context of RS image analysis for crop classification?
>> Authors should add the parameters of the algorithms.
>> A comparison with state of art in the form of table should be added
>> Results need more explanations. Additional analysis is required at each experiment to show the its main purpose.
>> The Limitations of the proposed study need to be discussed before conclusion.
>> Rewrite the Conclusion section to be:
- You must more clearly highlight the theoretical and practical implications of your research
-Discuss research contributions.
-Indicate practical advantages (in at least one separate paragraph),
-discuss research limitations (at least one separate paragraph), and
-supply 2-3 solid and insightful future research suggestions.

---

## Round 0.2 · accepted · Accept

Revise the final version of language and formatting according to the Reviewer.

Reviewer 2 ·

Basic reporting

after the corrections - acceptable

Experimental design

after the corrections - acceptable

Validity of the findings

after the corrections - acceptable

Additional comments

after the corrections - acceptable

you only need to improve the punctuation in mathematical formulas in a few places

Reviewer 3 ·

Basic reporting

The authors have been addressed all my comments correctly. No more comments are required from my side. The current version can be published in the journal.

Experimental design

The authors have been addressed all my comments correctly. No more comments are required from my side. The current version can be published in the journal.

Validity of the findings

The authors have been addressed all my comments correctly. No more comments are required from my side. The current version can be published in the journal.

Additional comments

The authors have been addressed all my comments correctly. No more comments are required from my side. The current version can be published in the journal.